# RSE-YOLOv8: An Algorithm for Underwater Biological Target Detection

**DOI:** 10.3390/s24186030

**Published:** 2024-09-18

**Authors:** Peihang Song, Lei Zhao, Heng Li, Xiaojun Xue, Hui Liu

**Affiliations:** Faculty of Information Engineering and Automation, Kunming University of Science and Technology, Kunming 650504, China; 20222204179@stu.kust.edu.cn (P.S.);

**Keywords:** underwater detection, YOLOv8, SAConv, RFAConv, SPPF

## Abstract

Underwater target detection is of great significance in underwater ecological assessment and resource development. To better protect the environment and optimize the development of underwater resources, we propose a new underwater target detection model with several innovations based on the YOLOv8 framework. Firstly, the SAConv convolutional operation is introduced to redesign C2f, the core module of YOLOv8, to enhance the network’s feature extraction capability for targets of different scales. Secondly, we propose the RFESEConv convolution module instead of the conventional convolution operation in neural networks to cope with the degradation of image channel information in underwater images caused by light refraction and reflection. Finally, we propose an ESPPF module to further enhance the model’s multi-scale feature extraction efficiency. Simultaneously, the overall parameters of the model are reduced. Compared to the baseline model, the proposed one demonstrates superior advantages when deployed on underwater devices with limited computational resources. The experimental results show that we have achieved significant detection accuracy on the underwater dataset, with an mAP@50 of 78% and an mAP@50:95 of 43.4%. Both indicators are 2.1% higher compared to the baseline models. Additionally, the proposed model demonstrates superior performance on other datasets, showcasing its strong generalization capability and robustness. This research provides new ideas and methods for underwater target detection and holds important application value.

## 1. Introduction

The diversity of underwater organisms plays a key role in maintaining ecological balance. Abundant underwater organisms constitute a complex and stable ecosystem through the food chain and food web, in which each species plays a specific role in maintaining the stability and function of the ecosystem [1]. However, the unreasonable development of underwater resources and climate change have significantly threatened the stability of the underwater ecological environment. Fluctuations or the disappearance of a particular species may trigger a cascading effect on the entire ecosystem, potentially leading to the disruption of the ecological balance [2].

The advent of deep learning technology has introduced a novel approach to the field of image processing and pattern recognition, offering a promising avenue by addressing the intricate challenges of underwater target detection. Deep learning technology can analyze input images using pre-trained models to achieve target detection and classification, thereby improving the efficiency of underwater ecological environment assessment [3,4,5]. However, the unique physical and optical properties of the underwater environment, such as light scattering, absorption, reflection, and refraction, lead to varying degrees of information loss in different channels of underwater images [6]. Traditional target detection algorithms struggle to directly address underwater target detection tasks, and the specific challenges of the underwater environment impose limitations on the resources available for detection devices [5].

In view of the above problems, this paper focuses on the study of common marine benthic invertebrates and proposes an underwater target detection algorithm, providing a reference for deployment on underwater mobile devices:This paper introduces an efficient receptive field attention convolution that enables the new convolution to fully account for the importance of different receptive fields. This design addresses the issue of parameter sharing and enhances the model’s sensitivity to variations in underwater environmental channel information, thereby tackling the problem of underwater image distortion caused by light absorption and scattering. Additionally, the introduction of the Global Average Pooling (GAP) layer enables the model to better capture global information from the image, thereby improving the identification of key features.In this paper, the C2f module is redesigned to incorporate SAConv (switchable atrous convolution), which allows for the flexible adjustment of the atrous rate within the same network layer. This modification enables the network to observe features at different scales through convolutional kernels with two different dilation rates, thereby improving its ability to address the challenge of large scale differences in underwater targets.Compared to traditional spatial pyramid pooling, this paper proposes a new ESPPF structure that reduces the depth of spatial pyramid pooling while expanding the module’s width to enable parallel pooling. This approach enhances the efficiency of feature fusion and reduces the number of model parameters, which is of significant importance for underwater target detection applications and embedded devices.

The structure of this study is as follows. Section 2 explains the related algorithm architecture and the foundational work related to this research. Section 3 describes the specific improvements of the proposed RSE-YOLO algorithm. Section 4 introduces the details of the experimental platform and the research datasets, presents the experimental results, and analyzes these results to illustrate the innovations and practicality of the RSE-YOLO algorithm. Section 5 primarily analyzes the experimental data and briefly discusses the results. Section 6 summarizes the content of this research and outlines future research directions.

## 2. Related Works

Traditional underwater object detection methods typically rely on classical image processing and computer vision techniques. These methods have the advantage of lower computational complexity but require manually designed feature extraction and rule-based algorithms. Additionally, they demand significant domain knowledge, and model tuning is often complex and requires iterative experimentation and optimization. In the development of traditional algorithm-based underwater object detection technology, Fatan et al. employed edge detection techniques to extract cable edges from images and then classified the extracted edges using a multilayer perceptron (MLP) neural network and a support vector machine (SVM) [7]. The filtered edges were subsequently refined using morphological operators. Finally, the refined edges were input into the Hough transform for cable detection. This approach addresses issues of blurriness and low contrast in underwater images of cables. Mathias et al. proposed an automated underwater object detection method that combines deep learning with traditional image processing techniques. The approach utilizes a Convolutional Neural Network (CNN) as the primary deep learning method [8]. The CNN is employed to extract highlevel features from the images and perform object classification. Subsequently, techniques such as edge detection, morphological operations, and feature extraction are applied in preprocessing and post-processing stages to enhance the saliency and recognizability of the detected objects.

In the development history of deep learning-based object detection technology [9], existing algorithms can be broadly categorized into two types based on their workflow: two-stage object detection algorithms and single-stage object detection algorithms [10]. Among them, the two-stage target detection algorithm is represented by Fast R-CNN [11] and Faster R-CNN [12]. First, the candidate regions are formed, and then each region is classified and regressed. These algorithms have high accuracy, but the detection speed is slow. The single-stage algorithm, represented by YOLO [13] and SSD [14], transforms the target detection task into a single regression problem and does not require region proposals. Single-stage algorithms have the advantages of fast speed and lower computational complexities, making them more suitable for real-time detection sensors. Therefore, they are widely used in underwater target detection systems [15,16].

In order to address various challenges in underwater target detection tasks, the paper by Song et al. proposed a method for detecting underwater organisms by integrating image enhancement using MSRCR to address issues of low contrast and poor image quality caused by the underwater environment. This is combined with Mask R-CNN to accomplish the task of underwater target detection [17]. This method achieves a high-precision detection of small sample datasets by combining image enhancement algorithms. Seese et al. proposed a fish detection and classification algorithm based on convolutional neural networks (CNNs). The algorithm first uses fusion technology to estimate the background image and perform foreground segmentation to identify the fish regions, thereby extracting the fish foreground from the complex background. Subsequently, deep convolutional neural networks (DCNNs) are applied to extract features and classify the segmented fish foreground regions [18]. Cai et al. proposed a collaborative weakly supervised learning method for underwater object detection that combines multiple weak supervision signals. This approach improves detection performance by optimizing and integrating weak label information from different sources [19]. An advanced lightweight two-stage model for underwater structural damage detection was proposed to enhance detection efficiency. The model addresses issues such as blurriness, fogging, distortion, and low resolution caused by light absorption and water flow by employing image enhancement techniques that improve detection accuracy [20].

Based on the single-stage detection framework, Hu et al. proposed a sea urchin detection network using the SSD algorithm [21]. To address the specific challenge of detecting the spiny edge features of sea urchins, they incorporated a multi-directional edge detection algorithm. This approach enhances the detection of detailed edge features, thereby improving the network’s capacity to accurately recognize sea urchins, which are distinguished by their complex and pronounced edge characteristics. Guo et al. proposed an underwater target detection method based on the YOLOv3 model incorporating MSRCP image enhancement techniques to address the issues of blurred and low-contrast underwater images [22]. Zhang et al. proposed a model to enhance the detection capabilities for small underwater targets. The model incorporates an attention mechanism to focus on critical features in the image, thereby improving the accuracy of detecting small underwater targets. Additionally, it integrates a multi-scale fusion strategy to capture information across different scales, further enhancing the recognition of small targets [23]. Hao et al. introduced the EASPP module into the backbone network of YOLOv4 [24]. Zhou et al. proposed an underwater optical detection network based on the YOLOv8 framework, which extracts more target features through the cross-stage multi-branch module and the large kernel spatial pyramid module to address the issue of poor image quality [25]. Zhang et al. proposed a precise method for fish detection in natural underwater environments named BSSFISH-YOLOv8. This method replaces the original convolution module with an SPD-Conv module to reduce the loss of fine-grained information and adds a 160 × 160 small target detection layer to improve sensitivity to smaller targets [26]. Li et al. proposed an improved underwater fish detection method based on YOLOv5. This method uses the Res2Net residual structure to represent multi-scale features with finer granularity, increasing the receptive field of the network while reducing the model’s computational requirements [4]. Tan et al. proposed an improved polarization image fusion method that enhances the contrast and detail of targets through the weighted image fusion of images captured at different polarization angles [27]. This method effectively distinguishes targets from the background in complex environments and is particularly significant for underwater target detection tasks. Deep learning-based object detection algorithms do not rely on domain-specific knowledge or manual feature extraction and are particularly adept at handling complex and variable image scenes. With advancements in hardware platforms, the training speed of deep learning models has significantly increased [28]. This paper focuses on single-stage object detection algorithms based on deep learning and examines the trade-offs between accuracy and model size. It differs from previous methods that combine image enhancement with underwater object detection, instead emphasizing the differences in information loss across various underwater channels and the size variations of marine invertebrate benthic organisms.

## 3. Materials and Methods

### 3.1. Materials

#### 3.1.1. YOLOv8 Detection Algorithm

YOLOv8 has been significantly improved based on YOLOv5. Compared to YOLOv5, the primary difference in YOLOv8 is the replacement of the C3 module in the backbone with the C2f module. By combining multiple branches and residual units, a gradient shunt connection is introduced in the new C2f module to enhance information flow in the feature extraction network while maintaining a lightweight design. The neck component primarily handles feature fusion. Unlike YOLOv5, which performs down-channel operations using 1 × 1 convolutions following up-sampling, YOLOv8 removes this operation, resulting in a more efficient and direct feature fusion process. This optimization in the neck improves information transfer efficiency and enhances the handling of features at different scales [29,30]. The head component is responsible for regression predictions. YOLOv8 adopts the advanced Task-Aligned Assigner strategy for sample allocation. This strategy improves sample matching during training and enhances the alignment and accuracy of detection tasks. In addition, the Task-Aligned Assigner sample allocation strategy is adopted in the sample matching process [31].

These improvements in YOLOv8 not only enhance its capability to process and analyze images but also streamline the overall detection pipeline. The optimized backbone, neck, and head structures perform exceptionally well in practical applications, excelling in both accuracy and processing speed. We can see the detailed structure of YOLOv8 in Figure 1.

#### 3.1.2. RFAConv Module

To enhance the network’s sensitivity to features in the receptive field and address the parameter sharing issue of convolution kernels, Zhang developed a novel mechanism known as Receptive Field Attention (RFA) [32]. This method effectively addresses the parameter sharing issue in convolution kernels and considers the importance of each feature within the receptive field, as shown in Figure 2.

Receptive Field Attention Convolution (RFAConv) is a convolution operation that combines group convolutions with an attention mechanism. It utilizes a 3×3 convolution kernel to extract features, with each window capturing spatial information within the receptive field. By effectively performing group convolutions, RFAConv can quickly extract and transform features within the receptive field. AvgPool implements global information aggregation in each receptive field, and the interaction between information blocks is completed by 1×1 group convolutions. Finally, the softmax function is used to emphasize the importance of each feature within its respective receptive field. The calculation of RFA can be expressed by Formula (Equation 1). RFAConv, developed by receptive field attention, can replace the standard convolution operation, effectively reducing computation cost and parameter increase, while improving network performance.
(1)$$\begin{array}{cc} \; & F=Softmax\left(g^{1\times 1}\left(AvgPool\left(X\right)\right)\right)\times RELU{\left(Norm\left(g^{k\times k}\left(X\right)\right)\right)}_{↔}\\ & =A_{rf}\times F_{rf}. \end{array}$$

#### 3.1.3. SAConv Module

In human vision, the selective enhancement and suppression of neuronal activation occur when observing objects. In the field of computer vision, Qiao et al. proposed the Switchable Atrous Convolution (SAC) mechanism, which achieves dual observations of the same feature. By introducing holes between convolution kernels (i.e., skipping certain pixels), atrous convolution expands the receptive field without increasing the number of parameters or computational load [33,34]. This improves computational efficiency. Switchable Atrous Convolution (SAC) is an advanced convolution mechanism that automatically selects different atrous rates [35]. It captures features at various scales by applying different atrous rates to the input features and incorporates a switching function using pooling and convolution layers. During inference, the switching component selects the appropriate atrous rate for the convolution operation based on the output value of the learned switching function S(x) from the training process, allowing the network to dynamically adjust the atrous rate of each convolution operation and aggregate the results.

#### 3.1.4. SPPF Module

The Spatial Pyramid Pooling Fast (SPPF) module utilizes Spatial Pyramid Pooling (SPP) techniques to aggregate feature information from various feature map levels, enabling the extraction of multi-scale features [36]. The SPPF module incorporates three max pooling layers that perform continuous pooling operations, as shown in Figure 3. These pooling layers aggregate feature information from different scales, effectively capturing multi-scale features. By pooling at different levels, the module enhances the network’s ability to recognize objects of varying sizes and positions. Its multi-scale feature extraction and efficient computation characteristics make it a crucial component in advanced object detection and image classification systems.

### 3.2. The Proposed RSE-YOLOv8

To enhance the accuracy and robustness of target detection in underwater environments, we developed a new algorithm called RSE-YOLO, as shown in Figure 4. The baseline model lacks sensitivity to variations in different receptive fields. To address this, we integrated the attention convolution module RFESEConv into the original backbone network, emphasizing contextual information to enhance feature extraction. Additionally, we designed a new feature extraction module, C2f_SAConv, which utilizes convolution kernels with two atrous rates to adapt to underwater targets of varying sizes. Finally, to address the high computational and parameter costs of traditional feature pyramids, the redesigned ESPPF module reduces the depth of the traditional feature pyramid while increasing the module’s width, thereby enhancing the accuracy of underwater target detection and reducing computational complexity.

#### 3.2.1. RFESEConv Module

To address the complexity of the underwater environment and the loss of channel information [37], we introduced the RFESEConv module into the YOLOv8 model. The specific structure of the RFESEConv module is shown in Figure 5. The RFESEConv module is an improvement based on the RFAConv module and includes a 3×3 convolution layer, an average pooling layer, a max pooling layer, and a 1×1 convolution layer. The input features are first subjected to global average pooling and a 1×1 convolution layer. The resulting features are then expanded to match the size of the original feature map and added element by element. The feature map then generates a channel attention map through the Effective-Squeeze-and-Excitation (ESE) attention module [38], and uses the fast grouping convolution method to extract the receptive field features. For each receptive field feature, global information is extracted using AvgPool. Finally, information interaction is accomplished through a series of 3×3 convolution operations. The importance of each receptive field feature is emphasized using the softmax function. The RFESEConv module enhances the traditional convolution module by improving the network’s spatial attention mechanism. Unlike the CBAM module [39], which separates the channel attention mechanism (CAM) [40] and the spatial attention mechanism (SAM) [41], the RFESEConv module employs a more efficient Effective-Squeeze-and-Excitation (ESE) channel attention mechanism to mitigate the impact of channel information loss differences in underwater environments. At the same time, both channel and spatial dimensions are weighted to enhance the model’s feature extraction capabilities and improve target detection accuracy in underwater images. This improvement not only reduces computational overhead but also allows each channel to obtain distinct attention maps, enhancing the model’s ability to capture target characteristics in the underwater environment. Additionally, pooling layers are introduced before and after the grouping convolution to enable the module to account for long-distance information and better address the issue of blurred target backgrounds in underwater environments.

In underwater environments, challenges arise in target detection due to light refraction, reflection, and low illumination, which degrade image quality. The RFESEConv module processes spatial feature information through grouping convolution and attention mechanisms and incorporates contextual information structures, thereby significantly enhancing the model’s feature extraction capabilities for underwater images and improving the accuracy and robustness of target detection.

#### 3.2.2. C2f_SAConv Module

In the field of computer vision, the primary task of an object detection model is to extract spatial information about objects from images using a convolutional neural network (CNN) [42,43], including their position and size. As the intermediate layer of the feature extraction network in the YOLOv8 model, the C2f module not only employs numerous standard convolution operations but also incorporates a bottleneck structure and cross-layer connections, thereby increasing the computational complexity of the model [44]. In underwater target detection tasks, detecting small targets in low-light conditions imposes higher accuracy requirements on the model [45].

To address these challenges, it is necessary to build a more efficient model structure. Enhancing the C2f module is an effective method (see Figure 6).

In this study, the SAC mechanism is introduced to redesign the bottleneck structure of the C2f module, thereby enhancing the model’s adaptability and performance. This design improves the network’s efficiency in processing complex images and is particularly effective in handling the significant scale variations of underwater objects. Additionally, the CBS component comprises a convolutional layer (Conv), batch normalization (BN), and the SILU activation function. To ensure the processed feature map retains its original size, a 1×1 convolution layer is employed to restore the output of the previous layer.

SAC_Bottleneck contains SAConv, CBS, and a 1×1 convolutional layer. The Switchable Atrous Convolution (SAC) components primarily include convolutions with atrous rates of 3 and 1, 5×5 pooling layers, and 1×1 convolution layers. The process of replacing standard convolution with SAConv is illustrated by Formula (Equation 2), where *x* represents the input, *w* denotes the weights, and *r* signifies the atrous rate. During the training phase, the model uses the same input features, producing two output feature maps of different scales through convolutions with atrous rates of 3 and 1. The input feature map simultaneously passes through a switching function composed of an average pooling layer and a 1×1 convolution layer to obtain a trainable S(x) value. In the inference stage, these two feature maps obtained by atrous convolution are multiplied by S(x) and 1−S(x), respectively. Finally, all the results are added to obtain an output feature map observed twice with different atrous rates.
(2)$$\begin{array}{cc} \mathrm{Conv}(x,w,1) & \overset{\mathrm{Convert}}{\to }S\left(x\right)\cdot \mathrm{Conv}\left(x,w,1\right)+\left(1-S(x)\right)\cdot \mathrm{Conv}\left(x,w+\Delta w,r\right) \end{array}$$

The C2f_SAConv module integrates SAC and the bottleneck structure of YOLOv8. The module retains the bottleneck structure and cross-layer connections of the C2f module and dynamically selects different atrous rates through the switching function, which enhances the feature extraction capability and is especially suitable for underwater environments. Meanwhile, the introduction of atrous convolution reduces the number of parameters to some extent [35].

#### 3.2.3. ESPPF Module

The SPPF module exhibits excellent multi-scale feature extraction capabilities. However, it is noted that SPPF utilizes three max pooling layers for continuous pooling, aggregating the branch information from these pooling layers. This continuous pooling may cause the deepest pooling layer to lose detailed information, making the association between the deepest pooling layer and the original input more blurred [46]. Therefore, we redesigned the SPP module by using two parallel pooling groups instead of single-path pooling, while reducing the depth of pooling.

To enhance the efficiency of multi-scale features extraction in underwater target detection, we replace the SPPF module with the ESPPF module. The specific structure of the ESPPF module is shown in Figure 7. The ESPPF module consists of two CBS modules, a 13×13 pooling layer, two 9×9 pooling layers, and a 5×5 pooling layer. When inputting features, a CBS module is initially employed to extract preliminary features. Subsequently, the obtained feature maps are split into two parts along the channel dimension, with each part being processed through separate branches. One branch includes two 5×5 pooling layers, while the other branch first passes through a 13×13 pooling layer and then through a 9×9 pooling layer. The feature map from the first CBS module and the outputs of the four pooling layers are then combined using a concatenation operation. Finally, a CBS module is applied to facilitate channel-wise information exchange across all feature maps, resulting in the final output. The redesigned ESPPF module enhances detection accuracy and multi-scale feature extraction capabilities through cross-stage partial pooling. At the same time, the number of parameters and computations is reduced, making it more advantageous compared to baseline models for deployment on underwater mobile and embedded devices.

## 4. Experimentation and Analysis

### 4.1. Experimental Dataset

This paper uses the publicly available underwater target detection dataset, UnderwaterBio (https://universe.roboflow.com/yolodetect-eh3nt/underwaterbio-d037p, accessed on 9 September 2024), to verify the effectiveness of the proposed model. The dataset includes four common benthic invertebrate organisms: sea urchins, starfish, conchs, and scallops. The targets appear in a variety of complex underwater environments, such as clear waters, turbid waters, and environments with significant light variation. The dataset also includes a substantial number of small and overlapping targets, reflecting key challenges in underwater target detection, such as low-light conditions, uneven distribution, poor data clarity, organism occlusion, partial concealment, and significant variations in target scale. In Figure 8, we present the statistics on the number of targets of various scales included in the dataset. We categorized and counted the number of objects based on the area of the ground truth boxes which ranges from smaller than 18 × 18 to greater than 90 × 90. We divided the dataset into a 7:2:1 ratio, resulting in a total of 3815 training samples, 1095 validation samples, and 544 test samples. These experimental results further demonstrate the superiority of our proposed improved model in terms of detection accuracy and generalization. In addition, we conducted a comparative experiment on the URPC2020 dataset (http://www.urpc.org.cn/index.html, accessed on 9 September 2024). URPC2020 is an underwater dataset similar to UnderwaterBio, comprising four categories: sea urchin, starfish, conch, and scallop. It includes a total of 5278 training samples, 1509 validation samples, and 754 test samples.

### 4.2. Experimental Settings

This study utilized the PyTorch 1.8.0 framework and the Ubuntu 22.04.2 operating system, with a program environment of Python 3.9 and CUDA 11.2. The training dataset was processed on a GeForce RTX 3090 24 GB GPU (NVIDIA, Santa Clara, CA, USA) and an Intel(R) Xeon(R) CPU E7-4809 v3 @ 2.00 GHz (Intel, Santa Clara, CA, USA). Each training session involved 200 iterations with a batch size of 4. The optimization algorithm used was SGD, with an initial learning rate of 1 × 10^−3^, a maximum learning rate of 1 × 10^−5^, a momentum of 0.937, and a weight decay of 5 × 10^−4^. The input image resolution was 640×640, and these training parameters and datasets were consistent across all models.

### 4.3. Evaluation Indicators

In target detection networks, the performance of the model is typically evaluated using the following metrics: precision, recall, mAP@0.5, mAP@0.5:0.95, speed (measured in frames per second or FPS), Params, and FLOPS [47,48,49]. Precision is the proportion of all positive samples predicted correctly in the model prediction results, where TP is the number of True Positives and FP is the number of False Positives. The calculation formula is shown in Formula (Equation 3).
(3)$$\mathrm{Precision}=\frac{\mathrm{TP}}{\mathrm{TP}+\mathrm{FP}}$$

Recall represents the proportion of correct predictions made by the model among all actual positive samples, where FN is the number of False Negatives. The calculation formula is shown in Formula (Equation 4):(4)$$\mathrm{Recall}=\frac{\mathrm{TP}}{\mathrm{TP}+\mathrm{FN}}$$

The mean average precision (mAP) represents the average value of average precision (AP) across all target categories. AP is the average precision at different recall rates. By calculating the precision–recall curve, the area under the curve is determined. The calculation of mAP can be expressed as:(5)$$\mathrm{AP}=\int_{0}^{1}P\left(R\right)dR;\mathrm{mAP}=\frac{\sum_{i=1}^{N_{\mathrm{cls}}}\mathrm{AP}}{N_{\mathrm{cls}}}$$
where *N* represents the number of classes, and ${\mathrm{AP}}_{i}$ denotes the average precision for class *i*. mAP@0.5 refers to the average AP with an IoU threshold of 0.5, where IoU measures the overlap between the predicted and ground truth boxes. It is computed as the intersection area of the two boxes divided by their union area. AP@0.5:0.95 evaluates performance using multiple IoU thresholds ranging from 0.5 to 0.95 with an interval of 0.05. This method allows for a more comprehensive assessment of the model’s performance under varying levels of positional accuracy, reflecting its ability to detect both high-precision (high IoU) and low-precision (low IoU) targets.

The number of parameters represents the total amount of all trainable parameters in the model. The number of parameters reflects the complexity of the model and is one of the most important indicators to measure the complexity of the deep learning model and the demand for computing resources.

The amount of floating-point operations is typically expressed in GigaFLOPS, where 1 GFLOPS equals $10^{9}$ floating-point operations. FLOPS are a critical measure of a model’s computational complexity and resource requirements. A higher FLOPS value indicates greater computational complexity and potentially longer inference times.

### 4.4. Ablation Experiment

To verify the effectiveness of the improvements proposed in this paper, we used YOLOv8n as the baseline model to analyze the independent effects of each module (C2f_SAConv, RFESEConv, and ESPPF). The experiment was conducted using the same dataset and training parameters to evaluate the model’s performance after removing each component or feature. The evaluation metrics include mAP@0.5, mAP@0.5:0.95, precision, recall, parameter quantity, and floating-point operations. Table 1 shows the performance comparison between the complete model and the ablation versions.

The experimental results show that the addition of the C2f_SAConv module increases the mAP@0.5 and mAP@0.5:0.95 by 1.2% and 1.1%, respectively, while the increase in the number of parameters and floating-point operations is minimal. After the RFESEConv module is added, mAP@0.5 and mAP@0.5:0.95 increase by 0.5% each. Although the number of parameters increases by 0.5 million, the floating-point operations are reduced by 1.7 GFLOPS. After adding the ESPPF module, mAP@0.5 and mAP@0.5:0.95 increased by 0.5% and 0.3%, respectively. At the same time, the parameters and floating-point operations were reduced by 0.13 million and 0.2 GFLOPS, respectively. The speed of the improved model was also enhanced compared to the baseline model. Overall, with the addition of each module, the accuracy of the model continued to improve, verifying the effectiveness of these modules. Although the number of parameters of the model increased by 0.57 million after all modules were added, the floating-point operations were reduced by 1 GFLOPS.

### 4.5. Contrast Experiment

To further verify the superiority and effectiveness of the improved algorithm proposed in this paper, comparative experiments with different types of target detection algorithms were conducted. The following algorithms were compared: YOLOv3-tiny, the lightweight version of YOLOv3 that introduces multi-scale prediction by predicting on three different feature map scales; YOLOv5, which utilizes the CSPNet (Cross-Stage Partial Network) structure and PANet (Path Aggregation Network) to enhance feature extraction and fusion, improve training speed, and reduce parameters and computational burden; YOLOv6, which employs RepConv (Re-parameterizable Convolution) technology to enhance feature fusion capabilities and improve detection accuracy, and YOLOv8, which introduces new backbone and neck designs to optimize feature extraction and fusion and adopts an improved Anchor-Free design for enhanced accuracy, speed, and flexibility. The results of different models in each evaluation index are shown in Table 2 and Table 3. The experimental results demonstrate that RSE-YOLOv8 performs exceptionally well across all evaluation metrics, particularly in terms of accuracy. Specifically, on the UnderwaterBio dataset, the mAP@0.5, mAP@0.5:0.95, precision, and recall of RSE-YOLOv8 improved by 2.1%, 2.1%, 1.2%, and 2%, respectively. To validate the model’s generalization ability, comparative experiments were conducted on the URPC2020 dataset. The results showed that the mAP@0.5, mAP@0.5:0.95, precision, and recall of RSE-YOLOv8 improved by 2.1%, 1.4%, 0.9%, and 1.5%, respectively. RSE-YOLOv8 also has only 3.5 million parameters and 7.1 GFLOPS of floating-point operations. Despite YOLOv5n and YOLOv8n having relatively few parameters, their overall accuracy remains lower than RSE-YOLOv8. YOLOv5s, while achieving high accuracy, has 9 million parameters and 23.9 GFLOPS, making it less suitable for underwater devices with limited computational resources. RSE-YOLOv8 exhibits a significantly lower number of floating-point operations, indicating excellent resource utilization and suitability for deployment on embedded systems or mobile devices.

### 4.6. Analysis of Improvement Effects

To evaluate the effect of the improved model, five groups of images from different scenes were tested. The size of the input image was set to 640 × 640, and the confidence threshold was set to 0.25. The experimental results are illustrated in Figure 9.

The first column shows the original image, the second column displays the detection results from YOLOv8, and the third column presents the results of the improved algorithm. The images in groups a and b show significant scale differences between the targets. YOLOv8 failed to detect both large and small targets, resulting in missed detections. RSE-YOLOv8 demonstrated better detection performance. The images in groups c and d illustrate cases with blurred images and complex backgrounds. YOLOv8 exhibited both misdetections and missed detections. RSE-YOLOv8 showed improved accuracy. The images in group e highlight the issue of numerous overlapping targets. RSE-YOLOv8 performed better at detecting overlapping targets compared to YOLOv8. The actual results demonstrate that RSE-YOLOv8 significantly enhances detection accuracy and reduces the missed detection rate in underwater target detection.

## 5. Discussion

The loss metrics for the improved model on the training and validation sets are as box_loss measures positioning loss by quantifying the difference between predicted and actual bounding boxes to optimize positioning accuracy; cls_loss evaluates the model’s classification accuracy, enhancing classification performance; and dfl_loss is used for bounding box regression [50], improving accuracy and stability by ac-counting for the shape of the predicted distribution. These curves Figure 10 reflect the trend of loss values on the training and validation sets. It is observed that, as training progresses, all losses exhibit a continuous decreasing trend, with the most significant reduction occurring in the first 50 epochs, indicating ongoing model optimization. The results show that all loss metrics stabilize over time, demonstrating the effective performance of the model.

Figure 11 illustrates the progression of key performance metrics for the RSE-YOLOv8 model over 200 epochs, including precision, recall, mAP_0.5, and mAP_0.5:0.95. The precision and recall curves exhibit a steady improvement, reflecting an enhancement in model reliability and robustness. The metric curves in the figure indicate a progressive increase in the model’s accuracy, particularly in its ability to detect objects across various scales. The gradual stabilization of these metrics suggests the convergence of the model’s learning process and its efficacy in achieving consistent performance.

Figure 12 visually illustrates the changes in the mAP metric of the improved algorithm and YOLOv8n during the training process. Although their accuracies are similar during the first 65 epochs, the improved algorithm’s accuracy significantly surpasses that of YOLOv8n after the 120th epoch, achieving a higher mean average precision upon convergence.

## 6. Conclusions

In this paper, we presented an enhanced YOLOv8 model, named RSE-YOLOv8, which can be used in underwater benthic invertebrate target detection tasks. By incorporating the C2f_SAConv, RFESEConv, and ESPPF modules, we substantially enhanced the model’s detection accuracy and robustness while keeping computational costs low. Experimental results demonstrated that the improved RSE-YOLOv8 outperforms the benchmark YOLOv8 across multiple evaluation metrics.

The RSE-YOLOv8 model was evaluated on two public underwater target detection datasets, UnderwaterBio and URPC2020. The results show significant improvements in mAP@0.5, mAP@0.5:0.95, precision, and recall. The RSE-YOLOv8 model maintains efficient feature extraction and target detection capabilities while keeping the number of parameters and computational requirements low. Specifically, the mAP and mAP@0.5:0.95 of RSE-YOLOv8 on the UnderwaterBio dataset increased by 2.1% and 2.1%, respectively, while on the URPC2020 dataset, they increased by 2.1% and 1.4%, respectively. Compared to other lightweight object detection models, RSE-YOLOv8 is more accurate and has an advantage over the benchmark model when deployed on mobile devices due to the small number of parameters and low computational requirements.

Through ablation experiments on each component of the improved model, we validated the contribution of each module to the overall performance enhancement. The C2f_SAConv module enhanced feature extraction capabilities by detecting objects at different scales using various atrous rates. The RFESEConv module improved detection accuracy while maintaining low computational cost. The ESPPF module optimized multi-scale feature extraction, reducing both parameters and computation, making the model better suited for detection tasks in complex underwater environments.

In summary, this study provides a novel approach for developing underwater benthic organism detection models and demonstrates potential application value in underwater detection robots and other resource-constrained mobile devices. The RSE-YOLO model offers valuable insights for the research of underwater benthic invertebrates. Although the RSE-YOLO model has shown progress, there is still room for further improvement, particularly in areas such as model robustness, generalization across diverse underwater conditions, and the handling of underwater image blur. Additionally, the current research does not address other marine organisms, such as swimming fish, coral reef health monitoring, and marine pollution detection and assessment. Future work will focus on these areas to enhance the model’s practical applicability and suitability for embedded devices. With the advancement of edge computing and hardware acceleration technologies, the real-time capabilities and efficiency of object detection technologies continue to improve. These technologies will enable the feasibility of real-time underwater monitoring systems. The official YOLOv8 documentation provides guidance on deploying models on embedded devices (https://docs.ultralytics.com/zh/guides/raspberry-pi/, accessed on 9 September 2024), facilitating the immediate processing and analysis of large volumes of video data. RSE-YOLOv8 is expected to play a crucial role in future underwater applications, advance underwater target detection technology, and provide robust technical support for marine scientific research and environmental protection.

## Figures and Tables

**Figure 1 sensors-24-06030-f001:**
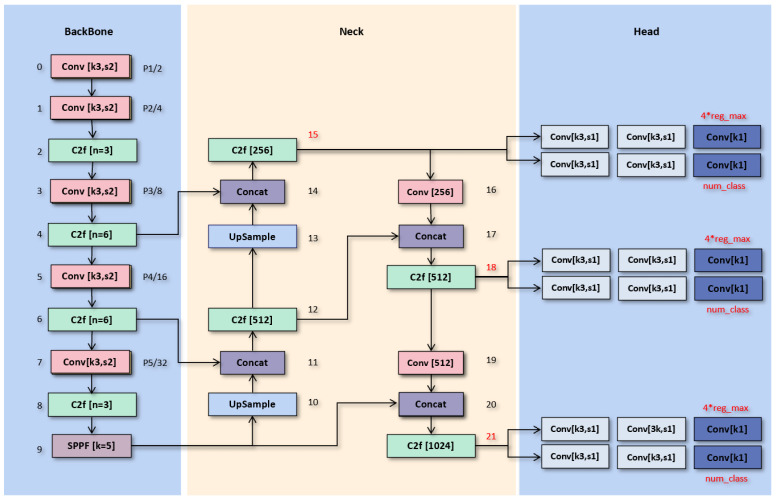
The detailed architecture diagram of YOLOv8.

**Figure 2 sensors-24-06030-f002:**
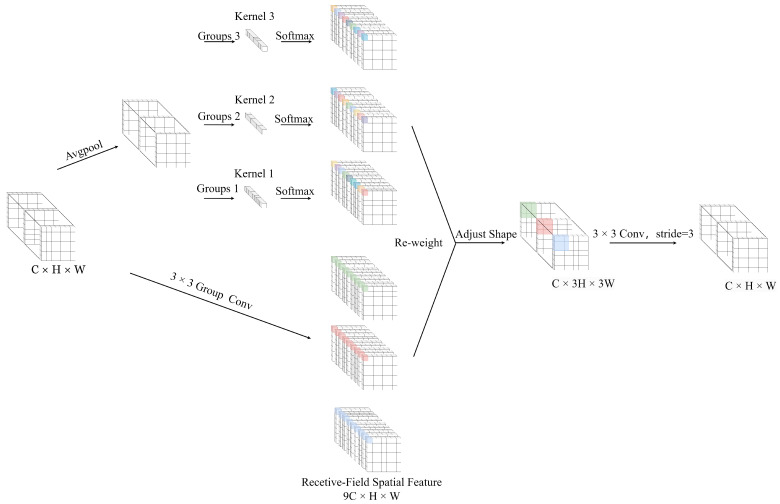
The detailed structure of RFAConv dynamically determines the importance of each feature within the receptive field and addresses the issue of parameter sharing.

**Figure 3 sensors-24-06030-f003:**
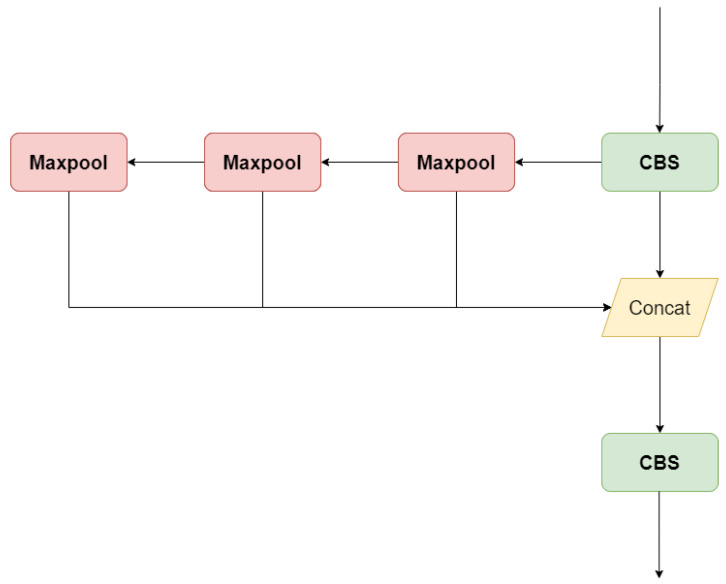
The detailed architecture diagram of SPPF.

**Figure 4 sensors-24-06030-f004:**
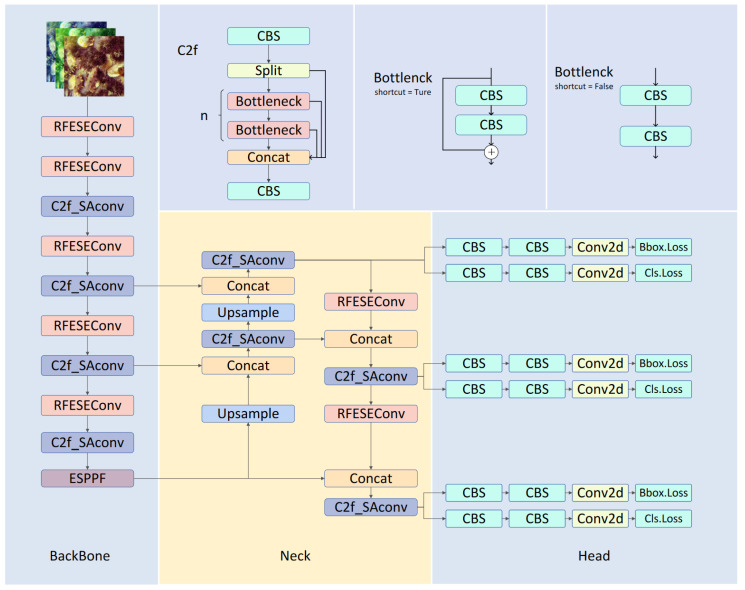
The detailed architecture diagram of RSE-YOLO primarily consists of three main components: the backbone, the neck, and the head.

**Figure 5 sensors-24-06030-f005:**
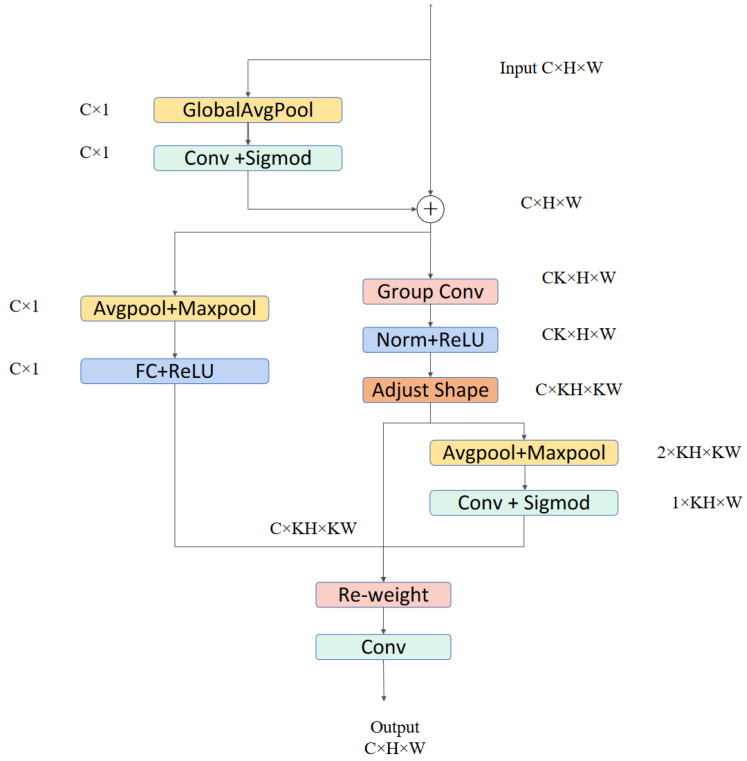
The detailed structure of RFESEConv.

**Figure 6 sensors-24-06030-f006:**
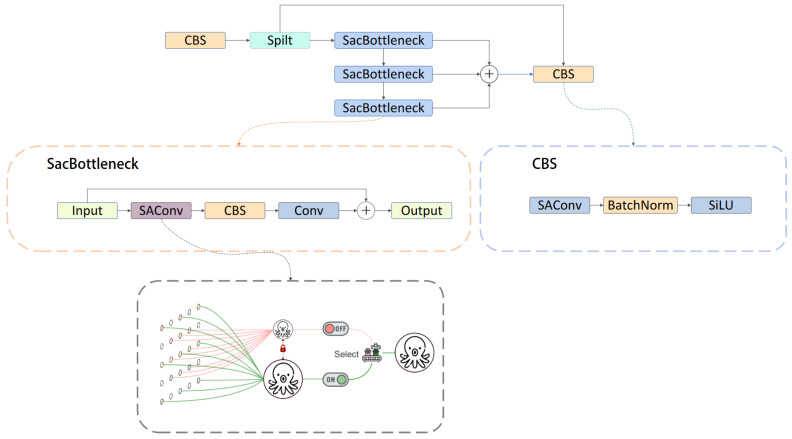
Structural diagram of C2f-SAConv.

**Figure 7 sensors-24-06030-f007:**
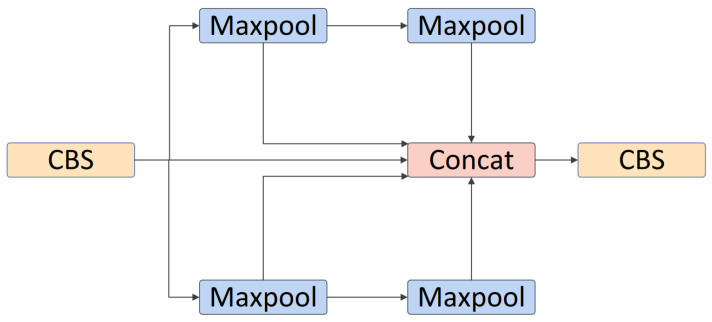
Structural diagram of ESPPF.

**Figure 8 sensors-24-06030-f008:**
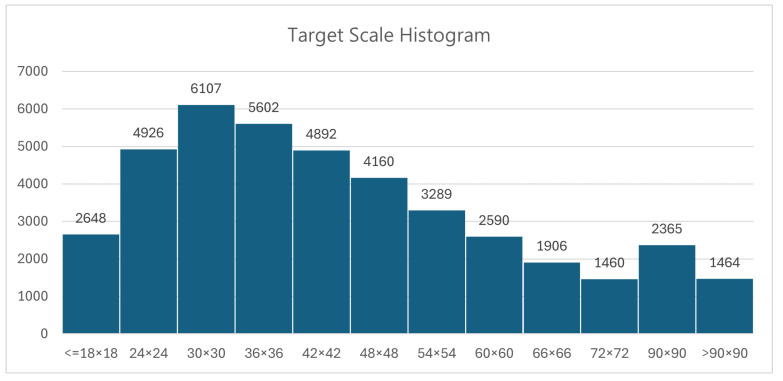
Statistics of the number of objects by different area sizes.

**Figure 9 sensors-24-06030-f009:**
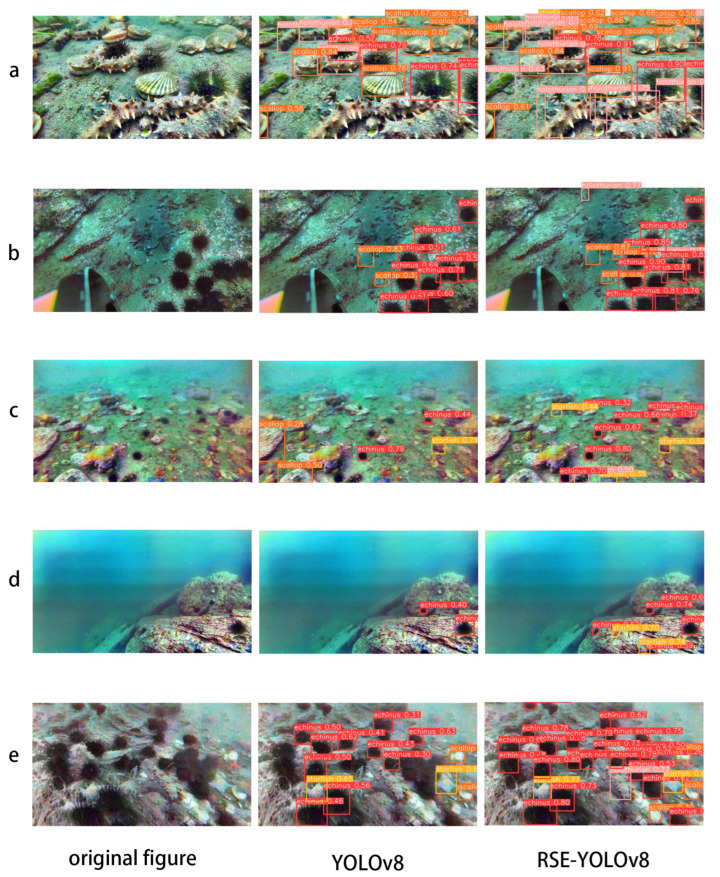
Visual results of the baseline model compared to the improved model. (**a**,**b**) represent images with significant differences in targets, (**c**,**d**) represent images with complex environments, and (**e**) represents images with a large amount of target overlap.

**Figure 10 sensors-24-06030-f010:**
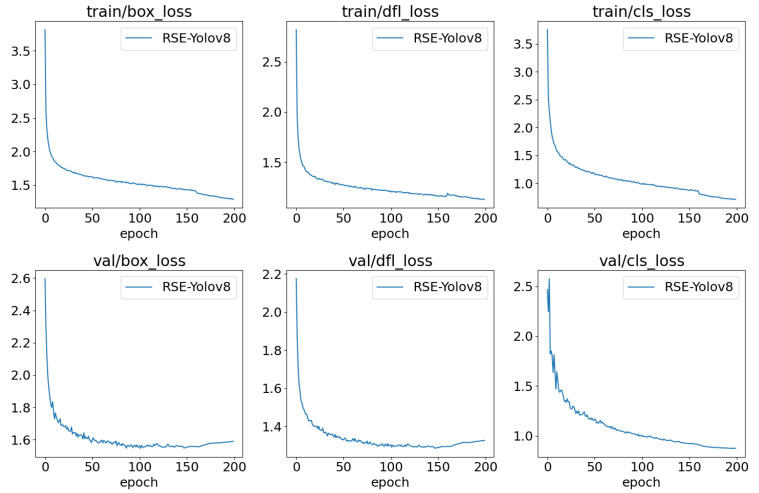
Loss curves for the training stages of RSE-YOLOv8.

**Figure 11 sensors-24-06030-f011:**
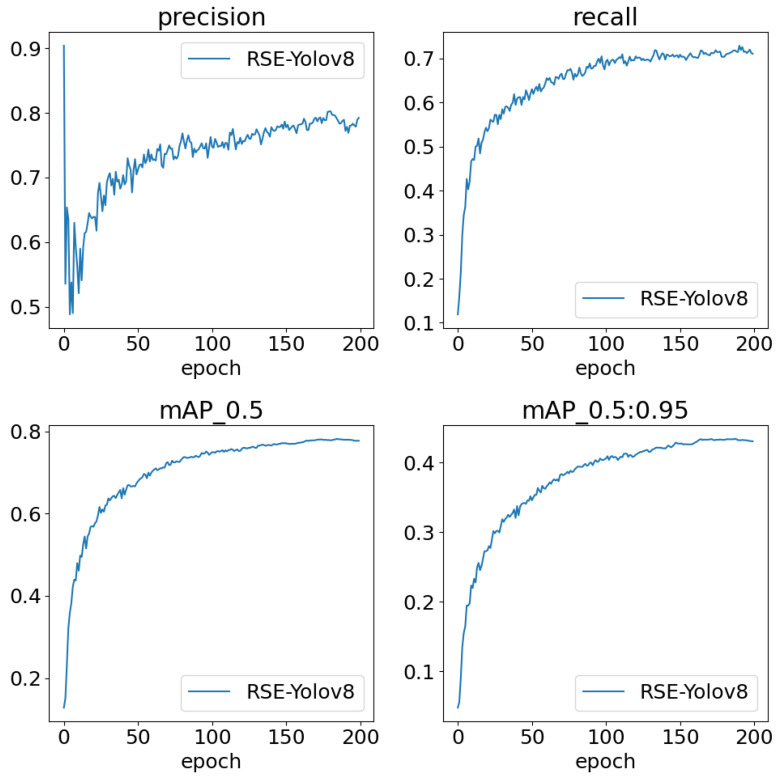
The metric variation during training for RSE-YOLOv8.

**Figure 12 sensors-24-06030-f012:**
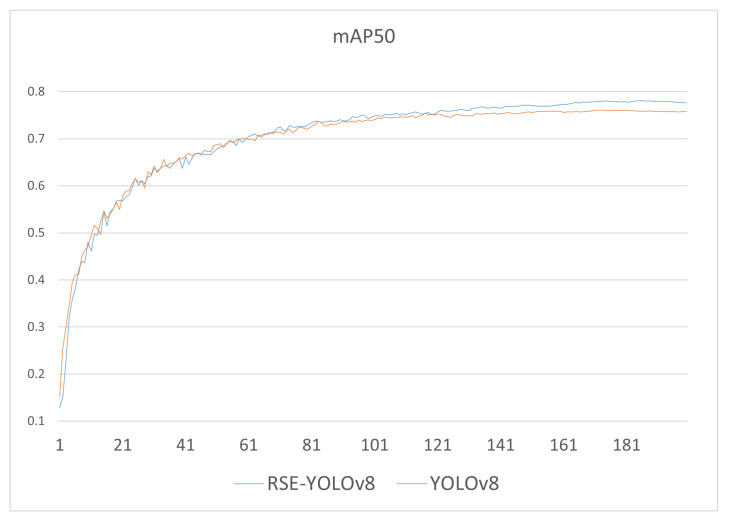
mAP@0.5 variation during training for YOLOv8 and RSE-YOLOv8.

**Table 1 sensors-24-06030-t001:** Experiment results of the ablation experiment.

YOLOv8n	C2f_SAConv	RFESEconv	ESPPF	mAP@0.5	mAP@0.5:0.95	Param (M)	FLOPS (G)	Speed
*√*				0.759	0.415	3.00	8.10	82.71
*√*	*√*			0.771	0.426	3.20	9.00	61.06
*√*	*√*	*√*		0.775	0.431	3.70	7.30	63.62
*√*	*√*	*√*	*√*	0.78	0.434	3.57	7.10	90.98

**Table 2 sensors-24-06030-t002:** Performance comparison of different object detection models in UnderwaterBio.

	Recall	Precision	mAP@0.5	mAP@0.5:0.95	Param (M)	FLOPS (G)
Faster-RCNN	0.489	0.37	0.616	0.267	82.3	25.1
SSD	0.793	0.48	0.600	0.276	13.5	15.2
YOLOv3-tiny	0.695	0.77	0.734	0.379	12.1	18.9
YOLOv5n	0.693	0.75	0.745	0.406	2.5	7.1
YOLOv5s	0.724	0.77	0.768	0.425	9.0	23.9
YOLOv8	0.698	0.78	0.759	0.415	3.0	8.1
YOLOV6	0.632	0.76	0.713	0.375	4.2	11.8
Ours	0.718	0.78	0.780	0.434	3.5	7.1

**Table 3 sensors-24-06030-t003:** Performance comparison of different object detection models in UPRC2020.

	Recall	Precision	mAP@0.5	mAP@0.5:0.95	Param (M)	FLOPS (G)
Faster-RCNN	0.564	0.376	0.690	0.325	82.3	25.1
SSD	0.638	0.540	0.676	0.337	13.5	15.2
YOLOv3-tiny	0.764	0.818	0.822	0.455	12.1	18.9
YOLOv5n	0.752	0.817	0.828	0.473	2.5	7.1
YOLOv5s	0.785	0.824	0.844	0.492	9.0	23.9
YOLOv8	0.763	0.816	0.832	0.484	3.0	8.10
YOLOv6	0.750	0.817	0.824	0.471	4.2	11.8
Ours	0.786	0.827	0.853	0.498	3.5	7.1

## Data Availability

The data are from public datasets, which are introduced in Section 4.1.

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
