# Peer review of "RSE-YOLOv8: An Algorithm for Underwater Biological Target Detection"

_sensors, 2024, doi:10.3390/s24186030_

Round 1
Reviewer 1 Report
Comments and Suggestions for Authors
There are many works for vision-based target detection of underwater cultural artifacts and benthic organisms. It is suggested to provide a more comprehensive summary of existing research on underwater target detection.
It is recommended that the ablation experiment section be supplemented with the inference time of the relevant algorithms on specific computing platforms, rather than just GFLOPs metrics, in order to provide a more intuitive demonstration of the effects of model compression.
Author Response
We sincerely appreciate the valuable critique and thoughtful suggestions provided by the reviewers. Based on these comments, we have thoroughly revised the manuscript. To facilitate identification, all changes in the revised version are highlighted in blue. Below, we address some of the questions raised.
We once again thank you for your feedback and constructive suggestions, which are invaluable in improving the quality of our manuscript
|
Comments 1: There are many works for vision-based target detection of underwater cultural artifacts and benthic organisms. It is suggested to provide a more comprehensive summary of existing research on underwater target detection. |
|
Response 1: We have supplemented the related work section by including additional information on traditional methods, such as references 7 and 8 in line 85, as well as methods combined with polarization imaging technology, such as reference 27 in line 175. We have also provided a summary at the end of the related work section.
|
|
Comments 2: It is recommended that the ablation experiment section be supplemented with the inference time of the relevant algorithms on specific computing platforms, rather than just GFLOPs metrics, in order to provide a more intuitive demonstration of the effects of model compression. |
|
Response 2: Thank you for your feedback. Table 1 in Section 4.4 (Ablation Studies) has been updated to include this metric. |
Reviewer 2 Report
Comments and Suggestions for Authors
This paper introduces an algorithm for underwater biological target detection based on RSE-YOLOv8. The experimental results show that the authors have achieved significant detection accuracy on the underwater dataset. This research provides new ideas and methods for underwater target detection and holds important application value. Some problems need to be improved:
1. The author needs to further clarify the innovation and contribution of this method. What is the biggest improvement or difference in this approach compared to existing methods?
2. For the most part, underwater images are blurry. In this case, how effective is this method? The author should discuss this part.
3. At present, a variety of image processing techniques have been developed. Polarization imaging technology also has great application potential underwater. You should also mention it in the literature review
4. The author needs to pay attention to some details in the paper, such as “table??” in line 345.
5. What underwater targets benefit most significantly from this technique? In what key application contexts does this method prove to be particularly advantageous?
6. In Figure 8, the text is too small; please adjust the font size and modify the spacing of the horizontal axis.
7.The author should enhance the description of the hardware to match the main thrust of the journal - sensors.
Comments on the Quality of English LanguageThe sentences in the article are relatively smooth, and there are no obvious grammatical problems.
Author Response
We sincerely appreciate the valuable critique and thoughtful suggestions provided by the reviewers. Based on these comments, we have thoroughly revised the manuscript. To facilitate identification, all changes in the revised version are highlighted in blue. Below, we address some of the questions raised.
We once again thank you for your feedback and constructive suggestions, which are invaluable in improving the quality of our manuscript
|
Comments 1: The author needs to further clarify the innovation and contribution of this method. What is the biggest improvement or difference in this approach compared to existing methods? |
|
Response 1: Thank you for the reviewer’s comments. We have added a brief comparison with existing methods at the end of Section 2, line 181, and summarized the contributions of this paper in the conclusion, as shown at line 593. |
|
Comments 2: For the most part, underwater images are blurry. In this case, how effective is this method? The author should discuss this part. Response 2: We apologize for the previous content of the article. This paper has made significant targeted improvements in underwater blur. We have revised the relevant description in line 35 of the introduction to enhance the rigor of the article and have added a prospect section on line 596 in the conclusion. Comments 3: At present, a variety of image processing techniques have been developed. Polarization imaging technology also has great application potential underwater. You should also mention it in the literature review. Response 3: Indeed, we have cited article 27 and introduced it in line 127 of the related work section. This article discusses the application of polarization imaging technology combined with object detection. Comments 4: The author needs to pay attention to some details in the paper, such as “table??” in line 345. Response 4: I apologize for this error. We have corrected the reference to this table, as shown in line 488. Comments 5: What underwater targets benefit most significantly from this technique? In what key application contexts does this method prove to be particularly advantageous? Response 5: In line 184 of Section 2, we clarified that our research subjects are based on invertebrate benthic organisms. Additionally, in the conclusion section, line 591, we briefly summarized the advantages of the model. Comments 6: In Figure 8, the text is too small; please adjust the font size and modify the spacing of the horizontal axis. Response 6: We have adjusted Figure 8, including changes to the text size and spacing, as shown in Figure 2 on page 18. Comments 7: The author should enhance the description of the hardware to match the main thrust of the journal - sensors. Response 7:We have increased the description related to hardware in the article, such as in line 42 of Section 1 and line 599 on page 20.
|
Reviewer 3 Report
Comments and Suggestions for Authors
The article is exciting and aligns with modern trends in computer vision development. However, it cannot be published in its current form. Please note the following shortcomings in this article:
Key notes:
1. It needs to be clarified why the authors focus on biological objects. According to the article, their model can recognize any underwater objects. The specifics of biology are not taken into account. For example, the needle-like nature of sea urchins, as noted in the work by the authors cited. Another feature of biological objects may be movement, for example, of plankton. Or movement in schools, for example, of fish. However, the authors only provide examples of stationary marine objects, such as shells.
2. The authors discuss the scalability of the objects their model can recognize. It would benefit them to establish specific limits, such as the most minor and significant objects that can be recognized. This helps define the model's applicability and its limitations clearly.
3. The authors provide two problems of underwater photography that they solve in their model - distortion and multi-scale. However, random disturbances and noise are a big problem underwater. The authors briefly mention them but need to provide unique suppressive methods. If their method does not require noise suppression and is noise-resistant, then it is necessary to provide experimental results proving this. First, we are talking about types of noise such as "salt and pepper" and blur.
4. The authors only make comparisons with neural network object recognition methods. While there are quite a lot of exciting works in the literature that provide classical methods of contour detection and further recognition, for example, based on Hough methods (for example, Zak, Bogdan & Garus, Jerzy. (2019). On a method for supporting visual identification of underwater objects. Polish Hyperbaric Research. 66. 25-34. 10.2478/phr-2019-0002.). There are also methods for recognizing multi-scale images, for example, based on the Gaussian pyramid (in https://persci.mit.edu/pub_pdfs/RCA84.pdf), without using neural networks. It is necessary to justify the advantages of the neural network method over this one.
5. In addition, there are studies in the literature that are very close to the topic of the authors' work - on improving Yolov8 for recognizing underwater living organisms - fish: Zheng, Tao & Wu, Junfeng & Kong, Han & Zhao, Haiyan & Qu, Boyu & Liu, Liang & Yu, Hong. (2024). A video object segmentation-based fish individual recognition method for underwater complex environments. Ecological Informatics. 82. 102689. 10.1016/j.ecoinf.2024.102689. Since this article is new, the authors may have yet to learn about it. However, it is necessary to analyze the differences between the approaches and prove the advantages of each solution.
6. The authors mention several times that their solution (the third of the highlighted problems) consumes little computing power and is suitable for use on mobile devices. However, as follows from Section 4.2 Experimental Settings, the authors required significant computing power for the experiments to train and test their model. Since the computing power required was significant, it is necessary to justify how their method can be used on low-power wearable and mobile devices.
7. The authors indicate SPEED (measured in frames per second or FPS) as one of the measures characterizing the efficiency of the proposed method. However, photographs, not video fragments, were used as datasets, so talking about "frames per second" seems inappropriate. Secondly, this value is not indicated anywhere in the article. The authors did not use this quality metric in their work.
8. The analysis of the results should be presented in a separate section «Discussion», distinct from the 'Experimentation and Analysis' section. This will help to maintain a clear and structured presentation of the research findings.
Design notes:
1. At the end of the Introduction section, a list of tasks that remain unsolved, despite all the described studies, should be formulated. Based on this, this article's main goal and objectives should be formulated.
2. Section 3.1. - YOLOv8 Detection Algorithm - is described unclearly. The model's description should not focus on technical features (sequence and combination of convolution layers, etc.) but describe the model from the point of view of the image processing process. At the same time, it is worth emphasizing those processing stages that, in the authors' opinion, can be improved. In addition, Figure 1, shown in the section, is not described in any way but has many designations, for example, "Spine-Neck-Head." The authors later provide these same elements in the description of their model but do not describe these parts in the original YOLOv8 model.
3. In the description section 3.2 of the proposed solution, The Proposed RSE-YOLOv8, each subsection (3.2.1. RFESEConv Module, 3.2.2. C2f_SAConv Module, 3.2.3. ESPPF Module) first provides theoretical background and references. This text is inappropriate here and should be moved to the appropriate sections on the literature review (2. Related Works) and/or theoretical background on previously developed models (as in section 3.1. YOLOv8 Detection Algorithm).
4. Section 4.7. - Analysis of THE Result - is extremely uninformative. If the authors want to provide illustrations of the accuracy of their method, they should present figures for all quality metrics used in the form of full-fledged diagrams.
Typos and carelessness:
1. Page 9, line 258. In the title "4. Experimentation and Analysist" the letter t at the end of the word Analysis should be removed.
2. Table 1 page 10. The first three columns should be rows. Also, the header for the YOLOv8n base method is skipped altogether.
3. Table 1, page 10. Last column. The heading GLOPS is indicated. Probably should read FLOPS.
4. Table 1 page 10. The table title is placed UNDER the table, not ABOVE it.
5. Page 11, line 345. The text contains an incorrect reference to a table. «in table ??».
6. Table 2, page 11. The last column heading is GFLOPS. Probably should be corrected to FLOPS.
7. Page 12 Fig. 7 should be presented more clearly, with the features of each column indicated at the outset rather than in the caption. These figures could be better presented as a table with three appropriately titled columns.
8. Section 5. Conclusions contains fragments that are repetitive in meaning and almost completely identical in spelling (lines 412 through 415 and 421 through 423, and others).
The authors should revise and submit the article to the journal editors again.
Comments on the Quality of English Language
The English language needs improvement
Author Response
We sincerely appreciate the valuable critique and thoughtful suggestions provided by the reviewers. Based on these comments, we have thoroughly revised the manuscript. To facilitate identification, all changes in the revised version are highlighted in blue. Below, we address some of the questions raised.
We once again thank you for your feedback and constructive suggestions, which are invaluable in improving the quality of our manuscript
|
Key notes Comments 1: It needs to be clarified why the authors focus on biological objects. According to the article, their model can recognize any underwater objects. The specifics of biology are not taken into account. For example, the needle-like nature of sea urchins, as noted in the work by the authors cited. Another feature of biological objects may be movement, for example, of plankton. Or movement in schools, for example, of fish. However, the authors only provide examples of stationary marine objects, such as shells.
|
|
Response 1: First, we sincerely thank the reviewer for their careful reading. We have revised the description of the research subjects in the article, focusing on the marine invertebrate benthic organisms included in the dataset. We have made modifications to the relevant expressions regarding the research subjects in the text, such as on line 43 of page 2 in Section 1, line 185 in Section 2, line 401 in Section 4.1, and lines 545, 591, and 593 in Section 6, to emphasize the research subjects of this study. |
|
Comments 2: The authors discuss the scalability of the objects their model can recognize. It would benefit them to establish specific limits, such as the most minor and significant objects that can be recognized. This helps define the model's applicability and its limitations clearly. |
|
Response 2: The scale issue addressed in this paper refers to the size of the objects to be detected within the image. Although this is related to the actual size of the biological objects, it does not definitively indicate that the actual size of the biological objects can be detected. Additionally, we conducted a statistical analysis of the objects of different sizes within the dataset, as shown in Figure 8 of Section 4, which demonstrates that the dataset contains objects of various scales. Moreover, around line 184 of Section 2, we clarified that our research subjects are based on invertebrate benthic organisms. |
|
Comments 3: The authors provide two problems of underwater photography that they solve in their model – distorion and multi-scale. However, random disturbances and noise are a big problem underwater. The authors briefly mention them but need to provide unique suppressive methods. If their method does not require noise suppression and is noise-resistant, then it is necessary to provide experimental results proving this. First, we are talking about types of noise such as "salt and pepper" and blur. |
|
Response 3: We apologize for the previously written article. This paper has not made any specific improvements in underwater noise suppression. We have revised the description in line 35 of the introduction section to improve the rigor of the article and have added a prospect on line 596 in the conclusion. |
|
Comments 4: The authors only make comparisons with neural network object recognition methods. While there are quite a lot of exciting works in the literature that provide classical methods of contour detection and further recognition, for example, based on Hough methods (for example, Zak, Bogdan & Garus, Jerzy. (2019). On a method for supporting visual identification of underwater objects. Polish Hyperbaric Research. 66. 25-34. 10.2478/phr-2019-0002 |
|
Response 4: We thank the reviewer for the provided references. Since it is not feasible to compare with traditional underwater target detection methods in a short time, we have cited related articles on underwater target detection, around line 177 in Section 2, to illustrate the benefits of neural networks. |
|
Comments 5: In addition, there are studies in the literature that are very close to the topic of the authors' work - on improving Yolov8 for recognizing underwater living organisms - fish: Zheng, Tao & Wu, Junfeng & Kong, Han & Zhao, Haiyan & Qu, Boyu & Liu, Liang & Yu, Hong. (2024). A video object segmentation-based fish individual recognition method for underwater complex environments. Ecological Informatics. 82. 102689. 10.1016/j.ecoinf.2024.102689 Response 5: We thank the reviewer for the provided references. Based on the research method of the article, we carefully reviewed the paper and found that the study differs from ours in terms of the dataset used. Our dataset does not include segmentation labels. Additionally, our research subjects—underwater benthic organisms—have more challenging edges to segment compared to fish. Moreover, the dataset in the cited literature contains more marine background negative samples, while our dataset’s negative samples mainly consist of seabed images, which have a more complex background environment and feature a large amount of object overlap. This makes it challenging to manually annotate segmentation labels. We will improve the related work section by providing the necessary description of previous methods and briefly summarizing it in the conclusion. Comments 6: The authors mention several times that their solution (the third of the highlighted problems) consumes little computing power and is suitable for use on mobile devices. However, as follows from Section 4.2 Experimental Settings, the authors required significant computing power for the experiments to train and test their model. Since the computing power required was significant, it is necessary to justify how their method can be used on low-power wearable and mobile devices. Response 6: In target detection models, training is usually conducted on large computing platforms to increase the training speed. The saved weight files are then deployed on mobile devices. Additionally, compared to the baseline model YOLOv8, our model requires fewer floating-point operations, which theoretically allows for faster inference speed when deployed on mobile devices. We have provided a method for porting the official model to a Raspberry Pi in the conclusion of the article. However, due to limitations in our current conditions, we do not have the appropriate equipment for deployment at this time. Comments 7: The authors indicate SPEED (measured in frames per second or FPS) as one of the measures characterizing the efficiency of the proposed method. However, photographs, not video fragments, were used as datasets, so talking about "frames per second" seems inappropriate. Secondly, this value is not indicated anywhere in the article. The authors did not use this quality metric in their work. Response 7: We appreciate the reviewer’s correction. In the field of deep learning target detection, datasets are usually provided in the format of image labels, and some datasets include video clips in the test set for evaluation. When performing inference on videos, it essentially involves processing the video frame by frame and inputting it into the model. We have already supplemented the speed metrics in Table 1 of Section 4.4 (Ablation Studies) based on the feedback from Reviewer 1 Comments 8: The analysis of the results should be presented in a separate section «Discussion», distinct from the 'Experimentation and Analysis' section. This will help to maintain a clear and structured presentation of the research findings. Response 8: Thank you for your suggestion. We have separated the analysis of the results into the "Discussion" section and expanded the content accordingly.
|
Design notes:
|
Comments 1: At the end of the Introduction section, a list of tasks that remain unsolved, despite all the described studies, should be formulated. Based on this, this article's main goal and objectives should be formulated. Typos and carelessness: |
|||||||||
|
Response 1: Thank you for your suggestion. The introduction briefly outlines the research content of the paper. We believe it may be more appropriate to address the unfinished tasks in the conclusion section, such as around line 594 of Section 6. |
|||||||||
|
Comments 2: Section 3.1. - YOLOv8 Detection Algorithm - is described unclearly. The model's description should not focus on technical features (sequence and combination of convolution layers, etc.) but describe the model from the point of view of the image processing process. At the same time, it is worth emphasizing those processing stages that, in the authors' opinion, can be improved. In addition, Figure 1, shown in the section, is not described in any way but has many designations, for example, "Spine-Neck-Head." The authors later provide these same elements in the description of their model but do not describe these parts in the original YOLOv8 model. |
|||||||||
|
Response 2: We have revised this part according to your suggestion, as described in Section 3.1.1. |
|||||||||
|
Comments 3: In the description section 3.2 of the proposed solution, The Proposed RSE-YOLOv8, each subsection (3.2.1. RFESEConv Module, 3.2.2. C2f_SAConv Module, 3.2.3. ESPPF Module) first provides theoretical background and references. This text is inappropriate here and should be moved to the appropriate sections on the literature review (2. Related Works) and/or theoretical background on previously developed models (as in section 3.1. YOLOv8 Detection Algorithm). Response 3: These contents are indeed more appropriately placed in the Materials section. As per your suggestion, we have moved these parts to Sections 3.1.2, 3.1.3, and 3.1.4. Comments 4: Section 4.7. - Analysis of THE Result - is extremely uninformative. If the authors want to provide illustrations of the accuracy of their method, they should present figures for all quality metrics used in the form of full-fledged diagrams. Response 4: Thank you for your suggestion. We have supplemented the presentation of the experimental results in Section 5, expanding the content related to the loss curves and accuracy curves.
Typos and carelessness:
|
Round 2
Reviewer 2 Report
Comments and Suggestions for Authors
1. Please try to keep the size of the text in the picture consistent.
2. The paper is substantial.
3. Table titles are not marked in Figure 8.
Comments on the Quality of English LanguageThe English quality of the paper can be further improved.
Author Response
|
Comments 1: Please try to keep the size of the text in the picture consistent. |
|
Response 1: Your suggestion is very good. We have adjusted the text size in the images to make them appear as consistent as possible, as seen in Figure 2 on page 5, Figure 3 on page 6, Figure 10 on page 15, and so on. |
|
Comments 2: The paper is substantial. |
|
Response 2: Thank you for the feedback. It is precisely because of your guidance and suggestions that the quality of this paper has improved. |
|
Comments 3: Table titles are not marked in Figure 8. |
|
Response 3: We have added a table caption to Figure 8 as requested. |
Reviewer 3 Report
Comments and Suggestions for Authors
I respect the authors for the great work they did to correct the article in such a short time. Although I still see some shortcomings, I give a positive conclusion about the possibility of publishing the article in the Sensors journal. I also hope that the authors will continue their research in the future and conduct new experiments that expand on these studies.
Author Response
|
Comments 1: I respect the authors for the great work they did to correct the article in such a short time. Although I still see some shortcomings, I give a positive conclusion about the possibility of publishing the article in the Sensors journal. I also hope that the authors will continue their research in the future and conduct new experiments that expand on these studies. |
|
Response 1: Thank you for your thorough review of our paper and for the valuable suggestions and guidance. The improvement in the quality of this article owes much to your significant contributions. We are grateful for your support and invaluable feedback. |